# Gelatin-Based Biofilms with Fe_x_O_y_-NPs Incorporated for Antioxidant and Antimicrobial Applications

**DOI:** 10.3390/ma15051966

**Published:** 2022-03-07

**Authors:** Johar Amin Ahmed Abdullah, Mercedes Jiménez-Rosado, Antonio Guerrero, Alberto Romero

**Affiliations:** 1Departamento de Ingeniería Química, Escuela Politécnica Superior, Universidad de Sevilla, 41011 Sevilla, Spain; mjimenez42@us.es (M.J.-R.); aguerrero@us.es (A.G.); 2Departamento de Ingeniería Química, Facultad de Física, Universidad de Sevilla, 41012 Sevilla, Spain; alromero@us.es

**Keywords:** biofilms, gelatin, nanoparticles, iron oxide, antioxidant activity, antibacterial activity

## Abstract

Currently, gelatin-based films are regarded as promising alternatives to non-environmentally friendly plastic films for food packaging. Nevertheless, although they have great biodegradability, their weak mechanical properties and high solubility limit their applications. In this way, the use of nanoparticles, such as Fe_x_O_y_-NPs, could improve the properties of gelatin-based biofilms. Thus, the main objective of this work was to include different concentrations of Fe_x_O_y_-NPs (0.25 and 1.0%) manufactured by green synthesis (GS) and chemical synthesis (CS) into gelatin-based biofilms in order to improve their properties. The results show that Fe_x_O_y_-NPs can be distributed throughout the biofilm, although with a greater concentration on the upper surface. In addition, the incorporation of Fe_x_O_y_-NPs into the biofilms improves their physicochemical, mechanical, morphological, and biological properties. Thus, it is possible to achieve suitable gelatin-based biofilms, which can be used in several applications, such as functional packaging in the food industry, antioxidant and antimicrobial additives in biomedical and pharmaceutical biomaterials, and in agricultural pesticides.

## 1. Introduction

A film is a very thin (thickness < 1 mm), transparent, and, in many cases, stretchable plastic with different uses [1]. It is normally made of polyethylene (PE) or polypropylene (PP), which offers great flexibility, making it perfect for wrapping products of different shapes and sizes [2]. The most interesting properties of these films are their transparency (which allows one to see inside the package), their flexibility and adaptation to all kinds of shapes and sizes, and their impermeability, which prevents the passage of air and moisture, acting as a barrier and protecting the interior [3]. These properties are very important in the food industry, as they help to extend the shelf life of food by preventing oxidation–reduction reactions and interaction with microbes [4].

In recent years, the film sector has been affected by an increase in demand due to two fundamental factors: the need for safety in the transport of products, thereby increasing the need for wrapping the merchandise, and the current trend towards better presented products, which has considerably increased the use of films as an element with hygienic and aesthetic characteristics [5]. Nevertheless, the low biodegradability of these packaging materials is generating a great pollution problem, being unsuitable for the food industry. In this way, natural biopolymer-based films are currently being investigated, such as those made of gelatin, chitosan, cellulose, or cellulose derivatives, which confer them good biodegradability without releasing toxic substances [6,7]. Among the raw materials, gelatin has great potential to be used in diverse industrial applications: (1) in the food industry, to protect food from certain factors such as drying, light, and oxidation, since it may be used in biofilms to incorporate a wide range of additives, including antioxidants, antimicrobials, antifungals, nutrients, and flavorings [8]; (2) in the cosmetics sector, e.g., in hair gels, shampoos, and other cosmetic products [9,10,11,12]; (3) in the biomedical sector, for intrinsic activities like antidiabetic, antihypertensive, anticancer, antimicrobial, and antioxidant activities, as well as in wound care and healing, tissue engineering, and gene therapy [13,14]; (4) in the pharmaceutical industry and medication delivery, e.g., as a gelling agent for plasma expanders, to manufacture soft and hard gelatin capsule shells, microencapsulation of pharmaceuticals and oils, emulsion stabilization, medicated sponges, scaffoldings, creams and gels, wound care, slow-release, and vaccinations [15,16]; (5) in the photography sector, as a protective coating that extends the life of photographs [12,17]; and (6) other applications such as paints and fertilizers [10]. All this is possible due to the fact that gelatin presents easy processability to form films, high flexibility, suitable gas barrier properties, high availability, and low cost [18,19]. Nevertheless, the greatest problem of gelatin-based biofilms is their high water solubility and vapor permeability, along with weak thermal and mechanical strength [20,21]. Many strategies have been suggested to overcome these problems. However, the most important strategy is the incorporation of nanomaterials as reinforcing fillers [22]. In this way, a variety of metal oxide nanoparticles, nanocellulose, and nanoclays have been incorporated into gelatin-based biofilms, such as zinc oxide nanoparticles [23], gelatin–silver NP antimicrobial composite films [24], titanium dioxide (TiO_2_-NPs) [25], gelatin biofilms reinforced with chitosan-NPs [26], montmorillonite [27], chitin NPs [28], and magnetic iron oxide NPs [29]. These nano-sized composites have been used for the manufacture of gelatin-based nanocomposite biofilms. In this sense, researchers are focusing their efforts on discovering alternatives to antibiotic feed additives that do not compromise productivity, since the usage of antibiotic feed additives has been a subject of increasing concern. Nanoparticles have recently been used to replace the high-cost organic source [30].

Furthermore, iron oxide nanoparticles (Fe_x_O_y_-NPs) have been widely used in biomedical applications due to their particular, unique, and magnetic characteristics, as well as their acceptable biocompatibility and bioavailability [31]. Fe_x_O_y_-NPs have a high inhibition capacity against the growth of different foodborne pathogens, such as *Staphylococcus aureus*, *Escherichia coli*, and *Pseudomonas aeruginosa* [32]. These NPs have proven to be capable of killing bacteria by producing reactive oxygen species (^•^OH, ^•^O_2_^−^), damaging bacterial DNA and proteins, leading to impaired mitochondrial function while keeping non-bacterial cells unharmed [2,18,20]. It is worth mentioning that Fe_x_O_y_-NPs are nonhazardous and non-cytotoxic at concentrations lower than 0.1 mg/mL [33]. Additionally, Faria et al. reported the oral therapeutic potential of iron oxide NPs to treat iron deficiency anemia [34]. In this context, Fe_x_O_y_-NPs are considered a suitable additive to incorporate into films and improve their antimicrobial activity.

Fe_x_O_y_-NPs can be synthesized by chemical or green approaches, which produce different characteristics. Chemically synthesized Fe_x_O_y_-NPs are more hazardous and have a higher tendency to agglomerate and to show lower stability. As a result of the introduction of green nanotechnology, researchers are focusing more on environmentally beneficial green or biological methods of producing Fe_x_O_y_-NPs. Thus, the nanoparticles synthesized by green methods are smaller, less agglomerated, more stable, and less toxic than those synthesized by chemical methods [35,36]. In addition, the green synthesis of these NPs can improve their purity and functional properties due to the high presence of active groups coming from polyphenols, which are used for their synthesis.

In this way, the main objective of this work was to develop gelatin-based biofilms with different concentrations of Fe_x_O_y_-NPs (0.25 and 1.0%). Green (GS) and chemical (CS) Fe_x_O_y_-NPs were used to compare their influence on the biofilm properties. To this end, the physicochemical, mechanical, microstructural, and functional properties of the different biofilms were evaluated.

## 2. Materials and Methods

### 2.1. Materials

The gelatin protein used in this study was food gelatin type B 200/220 g blooms supplied by Manuel Riesgo, S.A. (Madrid, Spain), being a food gelatin that contains sulfur dioxide (<10 ppm). Gallic acid (C_7_H_6_O_5_) and DPPH (2,2-diphenyl-1-picrylhydrazyl) were purchased from Sigma Aldrich (Darmstadt, Germany). All the reagents were of analytical grade.

Fe_x_O_y_-NPs were synthesized according to a previous work with slight modifications [37]. Briefly, it consists of colloidal precipitation in which 20 mL of *Phoenix dactylifera* L. extract, which is rich in polyphenols (green) or NaOH (chemical) (used as reductors) were mixed with 20 mL of FeCl_3_·6H_2_O (used as a precursor). The resulting 40 mL of mixture was heated under continuous stirring for 2 h at 50 °C. Then, the obtained precipitate was filtered, washed, and dried in an oven for 8 h at 100 °C. Finally, they were calcinated in a muffle for 5 h at 500 °C. 

CS Fe_x_O_y_-NPs had a mean size of 49 ± 2 nm, a 2.20 Fe_2_O_3_:Fe_3_O_4_ ratio, and 47% crystallinity. GS Fe_x_O_y_-NPs had a mean size of 32 ± 1 nm, a 0.84 Fe_2_O_3_:Fe_3_O_4_ ratio, and 69% crystallinity.

### 2.2. Biofilm Processing Method

Biofilms were fabricated by the casting procedure [3]. To this end, gelatin was firstly dissolved in distilled water (2% *w/v*), subjecting it to magnetic stirring for 2 h at 60 °C and 600 rpm. Subsequently, different concentrations of Fe_x_O_y_-NPs (0.25 and 1.0% *w/w*) with respect to the initial gelatin were dispersed in the solutions by ultrasound for 0.25 h. Finally, a constant volume (42.7 mL) of the solution was cast into Teflon plates (7.6 cm of diameter) and dried at room conditions (22 ± 1 °C and 35 ± 1% RH) for 3 days. The biofilms were peeled off and kept in a desiccator for further characterization. A reference biofilm was manufactured without the dispersion of NPs. Figure 1 shows a scheme of the different steps of this process.

### 2.3. Physicochemical Properties

#### 2.3.1. Water Solubility

For the determination of this parameter, the samples (2 × 2 cm^2^) were firstly weighted (*W_i_*) and then placed in an oven at 105 °C for 24 h. Later, the samples were immersed in 50 mL of distilled water for 24 h. Finally, the films were taken out and redried at 105 °C for 24 h to obtain the final dry weight (*W_f_*). The weight loss or water solubility percentage (*WS*%) was calculated with Equation (1) [26,29]:(1)$$WS\left(\%\right)=\frac{w_{i}-w_{f}}{w_{i}}\cdot 100$$

#### 2.3.2. Optical Properties

Another essential property of biofilms is their transparency, which restricts light transmission while allowing visibility through the packaging material. To determine this property, UV–vis spectroscopy was used. Thus, 1 × 2 cm^2^ samples were measured in a UV–vis spectrophotometer (Model 8451A, Hewlett Packard Co., Santa Clara, CA, USA) at 600 nm. A blank was carried out with air [24,38]. The results were indicated as transmittance (amount of light that can pass through the system).

### 2.4. Mechanical Properties

Static tensile tests were performed in order to evaluate the mechanical properties of the biofilms. For this, a modification of the ISO 527-3:2019 standard [39] was used. The samples were subjected to an increasing axial force at a speed of 10 mm/min until break in an MTS Insight 10 Universal Testing Machine (Berlin, Germany). During these tests, temperature and relative humidity were constant at 22 ± 1 °C and 35 ± 1%, respectively. The maximum stress (σ_max_), strain at break (ε_max_), and Young’s modulus of each biofilm were analyzed.

### 2.5. Morphological Properties

#### 2.5.1. Scanning Electron Microscopy (SEM)

Scanning electron microscopy (SEM) was used to determine the microstructure of the samples. The biofilms were analyzed on both sides (bottom and upper surfaces). In addition, their thickness was measured using ImageJ software. The samples were firstly covered with a thin gold layer to improve their conductivity and, thus, the quality of the micrographs. Later, they were observed in a Zeiss EVO microscope (Pleasanton, CA, USA) with an acceleration voltage of 10 kV and magnification of 3 KX [40].

#### 2.5.2. Energy Dispersive X-ray Spectroscopy (EDX)

The elemental composition distribution of the biofilms was also evaluated. For this, a EDX detector was attached to an SEM microscope [41]. In this way, the Fe distribution was analyzed in both surfaces of each biofilm through mapping (by converting to 8-bit grayscale and then Auto Threshold, where the obtained percentage was representative of the color area of Fe).

### 2.6. Functional Properties

#### 2.6.1. Antioxidant Activity

Antioxidant activity of the biofilms was determined using the protocol described by Mehmood et al. (2020) with slight modifications [29]. Thus, 1 mL of film solution was mixed with 1 mL of DPPH solution dissolved in methanol (40 ppm). This mixture was kept in the dark for 30 min at 25 °C. Finally, the absorbance of each solution was read at 517 nm in a spectrophotometer. Gallic acid was used as the positive control. DPPH inhibition (*IP*) was calculated using Equation (2).
(2)$$IP\;\left(\%\right)=\left(\frac{A-B}{A}\right)\times 100$$
where *A* and *B* are the DPPH absorbance without and with antioxidant agent, respectively.

#### 2.6.2. Antimicrobial Activity

The antimicrobial activity of the different biofilms was evaluated using an agar diffusion experiment [29]. In this way, cylindrical biofilms (9 mm of diameter) were firstly sterilized by immersion in 96% (*v/v*) ethanol for 2 min, after which they were rinsed thrice with sterile phosphate buffered saline (PBS). Then, they were placed in agar gels inoculated with *Staphylococcus aureus* (S. au) and *Escherichia coli* (E. col). The antibacterial activity was determined as the inhibition zone (diameters) surrounding the biofilm after 24, 48, and 72 h of incubation at 37 °C using ImageJ software.

### 2.7. Statistical Analysis

At least 3 replicates of each sample were performed for each measurement. The results were presented as mean values with standard deviations, which were calculated using IBM SPSS statistic software. In addition, the significant differences were evaluated using a one-way ANOVA with 95% confidence level (*p* < 0.05).

## 3. Results

### 3.1. Physicochemical Properties

#### 3.1.1. Water Solubility

Water solubility (WS) is a critical parameter for food packaging applications. In this sense, biofilms must be insoluble in water to improve water resistance and product safety [26,29]. Table 1 shows the water solubility (WS) values of the different biofilms. As can be seen, the reference biofilm (neat gelatin) reached the highest WS value (80.9%). Higher WS of neat gelatin biofilms was due to the hydrophilic nature of gelatin [42]. Thus, the incorporation of Fe_x_O_y_-NPs into the system improved its water resistance. Nevertheless, the increment in the Fe_x_O_y_-NP concentration did not significantly improve their water resistance. Therefore, it could be concluded that the incorporation of NPs could decrease the solubility in water, regardless of the incorporated concentration. These results could be due to the formation of strong hydrogen bonds between the biopolymer chains and NPs, as has already been reported in previous works [42]. Likewise, Wongphan et al. (2022) reported that the incorporation of enzymes into polymers could cause an interaction via hydrogen bonding that enhances the hydrophobic groups by reducing the contrast, which results in a decrease in the solubility [43]. These results are similar for GS and CS NPs.

#### 3.1.2. Optical Properties

The transmittance of each biofilm is also presented in Table 1. The reference system (neat gelatin) had 60% transmittance, which showed the great transparency of these biofilms. The incorporation of Fe_x_O_y_-NPs into gelatin-based biofilms reduced the transmittance values of the systems. Thus, this decrease was noted linearly with the increase of Fe_x_O_y_-NP concentration. Moreover, GS NPs led to a lower reduction than CS NPs, although this difference vanished at the highest NP content. The decrease in transparency brought about by NPs could be associated with the increase in solid material, which hindered the mobility of the biopolymer chains. Thus, the dispersion of fillers filled up free space preventing the passage of the light through it. Similar results were obtained in other works with gelatin-based biofilms containing silver or magnetic iron oxide NPs [24,29,44,45].

### 3.2. Mechanical Properties

Figure 2 shows the tensile profile of the different biofilms. As can be observed, the neat gelatin system presented a short elastic zone followed by a plastic one. This plastic zone decreased as the amount of Fe_x_O_y_-NPs in the biofilm increased. The comparison of the different systems is better defined through the mechanical parameters shown in Table 1. In this way, the addition of the Fe_x_O_y_-NPs generated an increase in Young’s modulus and maximum stress (σ_max_), being more pronounced in GS Fe_x_O_y_-NPs than in CS Fe_x_O_y_-NPs. This result is due to the fact that the presence of immiscible particles caused non-homogeneous networks and reduced extensibility of the films [45]. In contrast, no significant differences were found between the incorporated GS and CS-NPs at the same concentration for the strain at break (ε_max_) of the different films studied. Thus, the incorporation of Fe_x_O_y_-NPs stiffened the biofilms, probably due to the increasing presence of solid material in the biofilms and the increase of their thickness [29]. The incorporation of GS and CS-NPs may reduce the cohesion between the polymer chains and result in the reduction of strain at break [46]. On the other hand, the GS Fe_x_O_y_-NPs achieved greater maximum stress and Young’s modulus than the CS Fe_x_O_y_-NPs. This result could be due to the size of the NPs. In this way, smaller NPs generated a better interconnection in the structure, which highlighted this effect [47]. Thus, the incorporation of NPs always improved the mechanical resistance of the biofilms due to the strong network between NPs and biopolymer chains.

Nevertheless, the increase in NP concentration showed much lower effects as compared to their mere incorporation. Thus, no significant increase in Young’s modulus could be observed for either of the two types of NPs, while the effect on $\sigma_{\max}$ depended on the NPs used. The maximum stress was reduced when increasing GS NP content or increased when CS was used. These contradictory results could be attributed to the occurrence of two opposite effects. On one hand, an increase in NPs may induce an enhancement of the mechanical properties of the biofilms by promoting NP–protein hydrogen bonding. On the other hand, it may also favor NP agglomeration, which may interfere with the formation of these NP–protein domain arrangements [29]. Likewise, aggregation of nanoparticles in biodegradable polymers was possible with increasing nanofiller concentrations, which may lead to changes in the mechanical and barrier properties of the films [48].

### 3.3. Morphological Properties

#### 3.3.1. Scanning Electron Microscopy (SEM)

Figure 3 shows the bottom and upper surface of the different biofilms. As can be seen, the biofilm without NP incorporation (neat gelatin) presented a smooth and homogeneous structure on both surfaces. On the other hand, the incorporation of NPs generated rough surfaces with fissures and holes. Thus, the incorporation of NPs seemed to alter the nanostructure of biofilms, being more evident in the case of green nanoparticles (GS Fe_x_O_y_-NPs). This effect could be due to the functional groups present in GS Fe_x_O_y_-NPs (as they come from the polyphenols used in their synthesis), which could alter the structure of the biopolymer chains in the biofilms by charge interaction [49]. Furthermore, Fe_x_O_y_-NPs seemed to form more aggregates, probably due to the increase in concentration [48]. This effect was more evident in CS Fe_x_O_y_-NPs among films, which may be due to the non-availability of stabilizing agents [32].

#### 3.3.2. Energy Dispersive X-ray Spectroscopy (EDX)

EDX mapping confirmed the purity of biofilms and the synthesized NPs, since only the elemental components of proteins (C, H, N, O) and NPs (Fe, O) were present in the analyses. On the other hand, the distribution of Fe in the biofilms was evaluated through the Fe distribution found in the EDX analyses. Figure 4 shows this distribution in both surfaces (bottom and upper). Nevertheless, the colored area in each image was calculated to improve the comparison between the systems (Table 2). As can be seen, there was a homogeneous distribution of NPs in both surfaces of the biofilms, since Fe could be observed in the whole images. Nevertheless, there was a higher NP concentration on the upper surface than on the bottom surface, being more evident in the higher NP percentage (1%). This behavior can be explained by the hydrophobicity of the NPs, which made them migrate to the surface where they presented less steric repulsion, as well as by their density, which made them float instead of sinking into the film solution [50]. Additionally, this was explained by the strengthened Van Der Waals forces as particle size and concentration increased or that larger particles would exhibit higher surface tension, causing them to settle on the surface [50]. On the other hand, a higher predisposition to the agglomerations or aggregations of NPs could be observed at higher NP concentrations, causing them to fall by gravity. Among the different NPs, CS NPs presented greater precipitate than GS NPs, possibly due to their larger size. In addition, the highest concentration of CS Fe_x_O_y_-NPs showed agglomeration of NPs in certain areas of the biofilm. This behavior was already reported by Rufus et al. (2017) and Hosseini et al. (2015), who attributed it to an entropic process [26,36].

### 3.4. Functional Properties

#### 3.4.1. Antioxidant Activity

As can be seen in Table 1, the incorporation of Fe_x_O_y_-NPs produced an increase in antioxidant activity. In addition, the higher the Fe_x_O_y_-NP concentration, the higher the antioxidant activity. Thus, these nanoparticles act as antioxidants, as was reported in similar articles [29,51]. Regarding the different nanoparticles, the GS Fe_x_O_y_-NPs presented higher DPPH inhibition than the CS Fe_x_O_y_-NPs. This behavior could be due to the greater amount of antioxidant groups in them, conferred during their synthesis by polyphenols. In this way, green synthesis did not only display advantages in the synthesis process by reducing toxic waste and lowering costs, but it also allowed the generation of nanoparticles that provided greater functionality in future applications. These results showed that the gelatin-based Fe_x_O_y_-NPs displayed higher antioxidant activity than others reported in the literature, such as those obtained by Zafar et al. (2020) using iron oxide nanoparticles (IONPs) [29], and Shiv et al. (2019) using melanin nanoparticles [7]. It is worth being mentioned that the *IP%* of the positive control (gallic acid) was from 94.9 to 98.4%.

#### 3.4.2. Antibacterial Activity

The inhibition area produced by the different biofilms is collected in Table 3 through its significant measure (diameter). An image of these inhibition areas can be observed in the Appendix A. As can be seen, the neat gelatin biofilms showed antibacterial activity against *Staphylococcus aureus*, since a small zone of inhibition could be observed for the first 24 h. This could be due to the presence of sulfur dioxide (<10 ppm), as it is food gelatin. Nevertheless, this effect was lost in time when the bacteria grew. In this way, neat gelatin only had the ability to act on the peptidoglycan layer of bacteria to generate their lysis. However, it did not show the ability to act on the lipid layer, thus it only affected gram-positive bacteria and not gram-negative bacteria [52]. The inclusion of Fe_x_O_y_-NPs improved the antibacterial activity of biofilms. In this way, higher inhibition areas could be observed, as well as an effect on both types of bacteria. Nevertheless, although the difference in nanoparticle concentration on each of the surfaces of the biofilms was different, the antibacterial activity did not seem to have significant differences on each of the surfaces, making it a fully functional material. Regarding the synthesis of NPs, the GS Fe_x_O_y_-NPs presented higher antibacterial activity than the CS Fe_x_O_y_-NPs. This could be due to the smaller size of the green NPs, which conferred the biofilms a better capacity to inhibit the replication of bacterial DNA [53,54], as well as the higher crystallinity shown by GS Fe_x_O_y_-NPs [30]. Higher crystallinity allowed the released of Fe^+2^/Fe^+3^ to collide with the negatively charged membranes of bacteria, destroying their protein structure and causing them to die [55]. The incorporated antimicrobial function into biopolymer films could be an efficient method to enhance the shelf-life extension capacity for biodegradable films [56]. On the other hand, all the samples showed a decrease in their inhibitory capacity over time, being more notable in CS Fe_x_O_y_-NPs.

## 4. Conclusions

The incorporation of Fe_x_O_y_ nanoparticles into gelatin-based biofilms improved their properties even at a low concentration. Thus, the 0.25% (*w*/*w*) led to a reduction in the water solubility and to an improvement in the antioxidant activity and mechanical properties. An increase in NPs concentration led to opaquer or less transparent biofilms with greater thickness, but not to a significant improvement in the other properties evaluated, which may be due to the tendency of the nanoparticles to agglomerate at higher concentrations caused by their hydrophobic nature. Therefore, their inclusion reduced the water solubility of the biofilms and improved their mechanical resistance and functional properties. Furthermore, the dispersion of the nanoparticles incorporated by the casting method led to their heterogeneous distribution with differences in the distribution in both upper and bottom surfaces of the biofilms. However, these differences did not alter the antibacterial properties.

Finally, the green nanoparticles (GS Fe_x_O_y_-NPs) showed better antioxidant and antibacterial activities than the CS Fe_x_O_y_-NPs, which must improve the added value of these biofilms. This antioxidant activity of (GS Fe_x_O_y_-NPs) highlighted high antioxidant activity (*PI%* = 84.7% compared to gallic acid as a standard with *PI%* = 96.65). A control on the distribution of the nanoparticles in the biofilms and the use of the green nanoparticles led to sustainable biofilms with a significant improvement of the functional properties. All this makes these biofilms highly applicable for use as functional packaging to preserve food, antioxidant and antimicrobial additives in multiple applications such as biomedical and pharmaceutical biomaterials, as well as in agricultural pesticides. Nevertheless, further characterization of these biopolymer films, including a biodegradability study, will be required in future works as a previous stage to their use in packaging applications.

## Figures and Tables

**Figure 1 materials-15-01966-f001:**
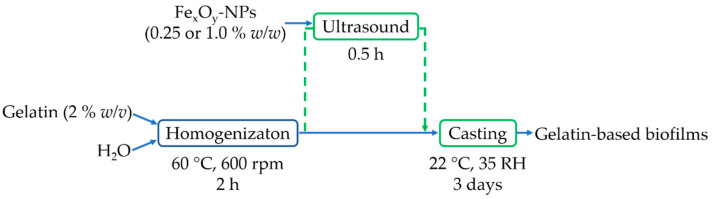
Scheme of the gelatin-based biofilm processing method.

**Figure 2 materials-15-01966-f002:**
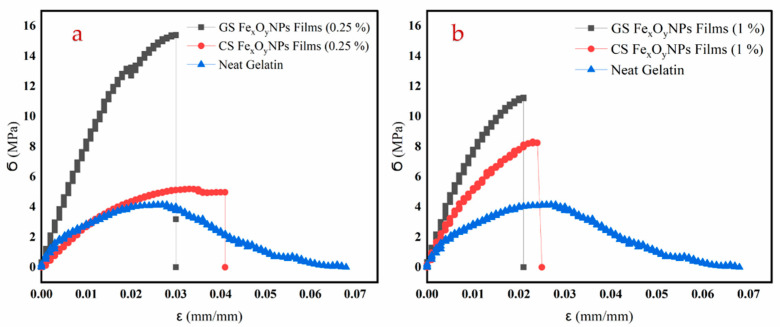
Tensile test profile of the biofilms with different concentrations ((**a**) 0.25 and (**b**) 1.0% *w/w*) of green (GS) and chemical (CS) Fe_x_O_y_ nanoparticles (NPs). Neat gelatin-based biofilm without NPs incorporated was used as the reference system.

**Figure 3 materials-15-01966-f003:**
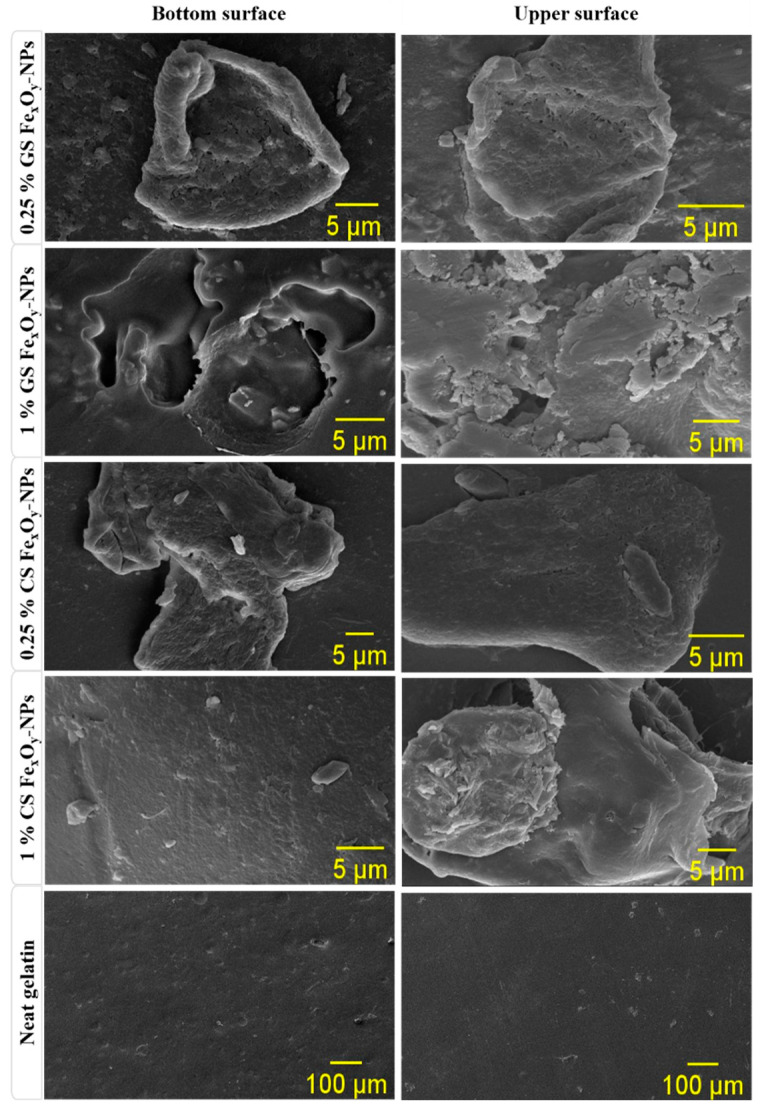
Scanning electron microscopy (SEM) images of the bottom and upper surfaces of the biofilms with different concentrations (0.25 and 1.0% *w/w*) of GS and CS Fe_x_O_y_-NPs. Neat gelatin-based biofilm without NPs incorporated was used as the reference system.

**Figure 4 materials-15-01966-f004:**
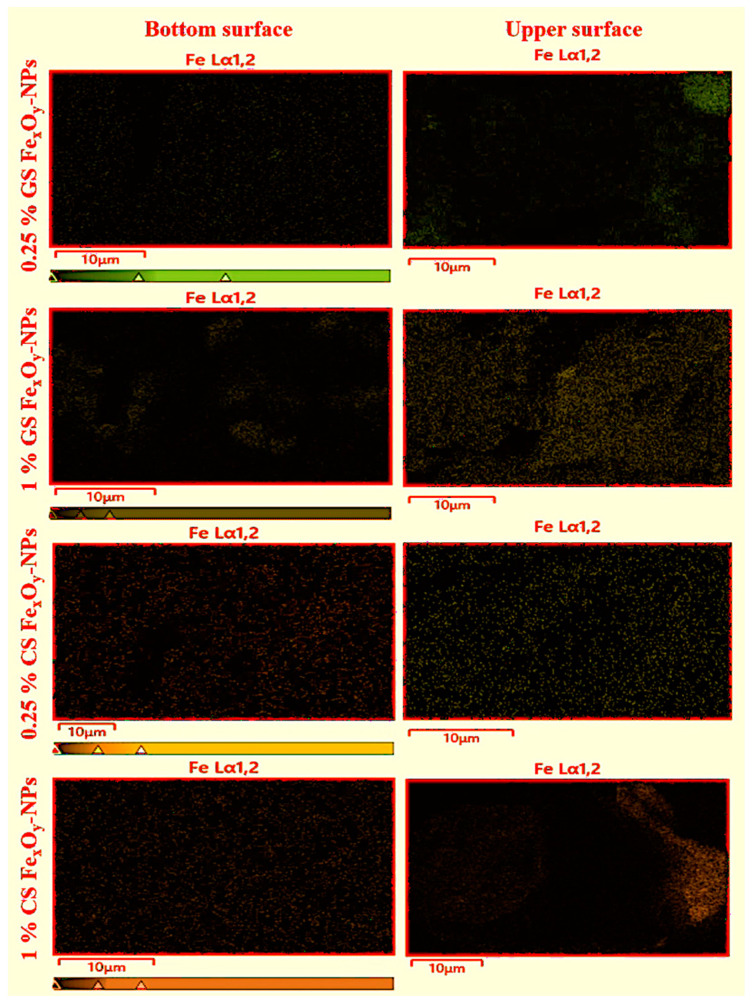
Fe distribution observed by energy dispersive X-ray spectroscopy (EDX) in the bottom and upper surfaces of the biofilms with different concentrations (0.25 and 1.0) of GS and CS Fe_x_O_y_ NPs. Neat gelatin-based biofilm without NPs incorporated was used as the reference system.

**Table 1 materials-15-01966-t001:** Physicochemical and mechanical parameters and antioxidant activity values of the biofilms processed with different percentages (0.25 and 1.0%) of green (GS) and chemical (CS) Fe_x_O_y_ nanoparticles (NPs). Neat gelatin-based biofilm without NPs incorporated was used as reference. Different superscript letters (a–d) of each column indicate significant differences (*p* < 0.05).

Sample	C% (*w*/*w*)	WS (%)	T_600_ (%)	Thickness (µm)	σ_max_ (MPa)	ε_max_ (mm/mm)	Young’s Modulus (MPa)	DPPH Inhibition (%)
GS Fe_x_O_y_-NPs	0.25%	66.8 ± 2.3 ^bc^	45.2 ± 0.3 ^b^	98.94 ± 0.8 ^c^	14.8 ± 0.8 ^a^	0.03 ± 0.01 ^a^	494.2 ± 8.9 ^a^	78.0 ± 1.9 ^ab^
	1%	64.1 ± 2.4 ^c^	29.2 ± 0.2 ^d^	105.6 ± 0.7 ^b^	11.9 ± 1.7 ^b^	0.02 ± 0.01 ^b^	586.6 ± 148.9 ^a^	84.7 ± 3.4 ^a^
CS Fe_x_O_y_-NPs	0.25%	69.6 ± 2.2 ^b^	42.0 ± 0.6 ^c^	104.9 ± 0.4 ^b^	5.2 ± 0.6 ^d^	0.03 ± 0.02 ^a^	193.9 ± 94.0 ^bc^	76.3 ± 2.7 ^b^
	1%	67.4 ± 1.5 ^bc^	29.3 ± 0.3 ^d^	109.1 ± 1.0 ^a^	8.4 ± 1.6 ^c^	0.02 ± 0.01 ^b^	263.5 ± 24.6 ^b^	79.6 ± 3.3 ^ab^
Neat Gelatin		80.9 ± 3.2 ^a^	60.0 ± 0.1 ^a^	89.2 ± 1.0 ^d^	4.6 ± 0.9 ^d^	0.07 ± 0.04 ^a^	67.3 ± 33.8 ^c^	44.5 ± 0.7 ^c^

**Table 2 materials-15-01966-t002:** Fe concentration (%) in the bottom and upper surfaces of the biofilms with different concentrations (0.25 and 1.0) of GS and CS Fe_x_O_y_ NPs.

Biofilms	Bottom	Upper
	Fe (%)	Fe (%)
GS Fe_x_O_y_-NPs 0.25%	12.0	16.5
GS Fe_x_O_y_-NPs 1.0%	13.3	30.3
CS Fe_x_O_y_-NPs 0.25%	12.9	13.6
CS Fe_x_O_y_-NPs 1.0%	16.7	43.4
Neat Gelatin	-	-

**Table 3 materials-15-01966-t003:** The inhibition areas (represented by their diameter in mm) produced by gelatin-based biofilm with 1.0% GS and CS Fe_x_O_y_-NPs incorporated against *Staphylococcus aureus* (S. au) and *Escherichia coli* (E. col). Neat gelatin-based biofilm without NPs incorporated was used as the reference system. Different superscript letters (a–e) in a column indicate significant differences (*p* < 0.05).

Test Time (h)	Biofilm	S. Au		E. Col	
		Upper	Bottom	Upper	Bottom
0		9 ^b^	9 ^c^	9 ^c^	9 ^b^
24	Neat Gelatin	9.4 ± 0.3 ^b^	9.4 ± 0.3 ^c^	0.0 ^e^	0.0 ^c^
	GS-NPs	12.8 ± 1.5 ^a^	13.5 ± 1.7 ^a^	13.4 ± 2.3 ^a^	15.5 ± 2.6 ^a^
	CS-NPs	12.9 ± 1.1 ^a^	11 ± 1.3 ^bc^	12.5 ± 2.0 ^ab^	9.3 ± 2.4 ^b^
48	Neat Gelatin	0.0 ^c^	0.0 ^d^	0.0 ^e^	0.0 ^c^
	GS-NPs	12.3 ± 2.3 ^a^	12.6 ± 1.6 ^ab^	11.1 ± 0.7 ^b^	14.6 ± 1.9 ^ab^
	CS-NPs	10.9 ± 2.9 ^ab^	10.9 ± 2.5 ^bc^	11.5 ± 0.4 ^ab^	0.0 ^c^
72	Neat Gelatin	0.0 ^c^	0.0 ^d^	0.0 ^e^	0.0 ^c^
	GS-NPs	9.1 ± 0 ^b^	9.5 ± 0.1 ^c^	6.1 ± 0.5 ^d^	8.3 ± 0.8 ^b^
	CS-NPs	0.0 ^c^	0.0 ^d^	6.7 ± 1.6 ^d^	0.0 ^c^

## Data Availability

The data presented in this study are available upon request from the corresponding author.

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
