# Peer review of "Gelatin-Based Biofilms with FexOy-NPs Incorporated for Antioxidant and Antimicrobial Applications"

_materials, 2022, doi:10.3390/ma15051966_

Round 1
Reviewer 1 Report
This manuscript has been resubmitted. Only few points have been revised as the recommended in the first review. Here are some points to revise.
L10 Please reconsider that the gelatin can be “substitutes” for PE and PP as the properties are much different.
L104 As there is no published work about the FeO preparation, more detail of the synthesis should be explained here.
L122-123 This should not be in the Materials and method section.
Were the films dried before measurement? If not, did the Wi include the weight of sorbed water?
Wf refers to weight after drying?
L191 If the data are insignificant difference, they should not be discussed as lower.
L194 Wongphan (2022 Food Packaging and Shelf Life) indicated interaction between incorporated enzymes and polymers via H-bonding which reduced numbers of hydrophilic groups in the polymer chains and decreased solubility.
L209 Why increased solid hindered mobility? Why/how the light transmission was affected by mobility?
L210 Dispersion of fillers filled up free volume which was unoccupied by the solids which the light pass through (Chatkitanan 2021 Meat Science).
L219 Presence of immiscible particles caused non-homogeneous networks and reduced extensibility of the films (Leelaphiwat 2022 Food Chemistry). Klinmalai (2021, LWT) indicated reduction of elongation due to reduced cohesion between the polymer chains.
L273 Phothisarattana (2021 Polymers) also indicated aggregations of nanoparticles in biodegradable polymers with increasing concentrations of nanofillers which subsequently affected mechanical and barrier properties of the films.
Fig. 4 Some pictures are too dark and the metal are not clearly observed. Please adjust and increase the brightness.
L299 Why does neat gelatin show antimicrobial activity? Does the inhibition area include diameter of the films?
L303 Revise English. What is “initial antibacterial activity…”?
L316 Why high crystallinity caused better microbial inhibition.
L314 Incorporated antimicrobial function into biopolymer films was efficient method to enhance the shelf-life extension capacity for biodegradable films (Laorenza 2021 Food Chemistry).
Author Response
Reviewer: 1
This manuscript has been resubmitted. Only few points have been revised as the recommended in the first review. Here are some points to revise.
- L10 Please reconsider that the gelatin can be “substitutes” for PE and PP as the properties are much different.
The authors apologize for the misunderstanding generated in this sentence. This wanted to express that the gelatin is a great candidate to substitute conventional plastics in general. These conventional plastics are used widely in the food industry. The problem of these packaging materials is not only their lower biodegradability which causes great pollution but they are also associated with limitations that could make changes in organoleptic characteristics. According to this comment, we have rewritten the first sentence of the abstract (L10-11) and included more information in the introduction section (L39-43).
- L104 As there is no published work about the FeO preparation, more detail of the synthesis should be explained here.
We agree with this comment, and we appreciate the reviewer for pointing out this problem. The protocol has been included in the “Materials and Methods” section (L106-112).
- L122-123 This should not be in the Materials and method section.
We thank this specification. This sentence has been included in results section (L192-194).
- Were the films dried before measurement? If not, did the Wi include the weight of sorbed water?
The authors would appreciate the query and we apologise for the lack of description in this section. We have considered this question and the details of measurement with the appropriate citations have been included (L130-135).
For the determination of this parameter, the samples (2 × 2 cm2) were firstly weighted (wi) and, then, placed in an oven at 105 ºC for 24 h. Later, the samples were immersed in 50 mL of distilled water for 24 h. Finally, the films were taken out and redried at 105 ºC for 24 h to obtain the final dry weight (wf). The weight loss or water solubility percentage (WS%) was calculated with Equation 1 (Andreuccetti et al. 2012; Hosseini et al. 2015; Mehmood, Sadiq, and Khan 2020): (1)
REFERENCES
Andreuccetti, C.; Carvalho, R.A.; Galicia-García, T.; Martinez-Bustos, F.; González-Nuñez, R.; Grosso, C.R.F. Functional properties of gelatin-based films containing Yucca schidigera extract produced via casting, extrusion and blown extrusion processes: A preliminary study. J. Food Eng. 2012, 113, 33–40, doi:10.1016/j.jfoodeng.2012.05.031.
Hosseini, S.F.; Rezaei, M.; Zandi, M.; Farahmandghavi, F. Fabrication of bio-nanocomposite films based on fish gelatin reinforced with chitosan nanoparticles. Food Hydrocoll. 2015, 44, 172–182, doi:10.1016/j.foodhyd.2014.09.004.
Mehmood, Z.; Sadiq, M.B.; Khan, M.R. Gelatin nanocomposite films incorporated with magnetic iron oxide nanoparticles for shelf life extension of grapes. 2020, doi:10.1111/jfs.12814.
- Wf refers to weight after redryg?
The Wf refers to the final weight after the redrying (L132). We hope that was clarified in the prior comment.
- L191 If the data are insignificant difference, they should not be discussed as lower.
The authors agree with the reviewer and the sentence has been changed (L197-201).
- L194 Wongphan (2022 Food Packaging and Shelf Life) indicated interaction between incorporated enzymes and polymers via H-bonding which reduced numbers of hydrophilic groups in the polymer chains and decreased solubility.
We appreciate the reviewer for sharing the useful data and enriching the manuscript. This reference has been included in the manuscript (L202-206).
- L209 Why increased solid hindered mobility? Why/how the light transmission was affected by mobility?
This is due to the fact that the fillers filled up the free space of the biofilms, preventing the passage of light through it. This discussion has been included in the manuscript (L220-224).
- L210 Dispersion of fillers filled up free volume which was unoccupied by the solids which the light pass through (Chatkitanan 2021 Meat Science).
This reference has been included to discuss the phenomenon explained in the previous question (L224).
- L219 Presence of immiscible particles caused non-homogeneous networks and reduced extensibility of the films (Leelaphiwat 2022 Food Chemistry). Klinmalai (2021, LWT) indicated reduction of elongation due to reduced cohesion between the polymer chains.
The authos acknoledge the information provided for the reviewer. We have considered these references and they have beeen used to discuss the mechanical properties founded in our systems L232-239.
- L273 Phothisarattana (2021 Polymers) also indicated aggregations of nanoparticles in biodegradable polymers with increasing concentrations of nanofillers which subsequently affected mechanical and barrier properties of the films.
We appreciate the continuous helpful suggestions. This discussion has been added in the manuscript (L253-256).
- 4 Some pictures are too dark and the metal are not clearly observed. Please adjust and increase the brightness.
Figure 4 has been improved according to the reviewer’s suggestion.
- L299 Why does neat gelatin show antimicrobial activity? Does the inhibition area include diameter of the films?
Gelatin presents a certain antimicrobial capacity, surely due to the incorporated additives (i.e. sulfur dioxide) as it is food gelatin. This supposition has been incorporated in the manuscript (L102-103 and L327-330).
- L303 Revise English. What is “initial antibacterial activity…”?
We apologise for the misunderstood description, and we appreciate the reviewer for this comment. According to this comment, we have considered rewriting the context (L327-330).
- L316 Why high crystallinity caused better microbial inhibition.
Higher crystallinity allowed the released Fe+2/Fe+3 to collide with the negatively charged membranes of bacteria, destroying their protein structure and causing them to die (Shuai et al. 2020).
This explanation has been included in the manuscript (L342-344).
- L314 Incorporated antimicrobial function into biopolymer films was efficient method to enhance the shelf-life extension capacity for biodegradable films (Laorenza 2021 Food Chemistry).
The authors acknowledge the reference provided by the reviewer. This reference has been incorporated in the manuscript (L346).

Reviewer 2 Report
In this manuscript, the authors reported gelatin-based biofilms fused with different concentrations of FexOy-NPs and evaluated its performance in antioxidant and antimicrobial applications. Overall the work is well designed and the research topic is of interest to certain community. Whereas more solid data are needed to support the conclusion and a few issues should be addressed before acceptance of publication.
- There are a lot of antioxidant and antimicrobial biofilms that were either commercialized or reported. What is the strength of this gelatin-based FexOy biofilms compared with them? And the strength should be highlighted in the manuscript.
- Biodegradability or the degradation behavior of this gelatin-based FexOy biofilms should be studied and presented.
- Besides tables, many results of biofilms should be presented in the form of graphs or figures. E.g., mechanical parameters, antioxidant activity in table 1, and antibacterial activity in table 3.
- There might be inconsistency between the color of pseudo-color scale bar and images in Figure 4. The authors should better double check.
- Maybe I missed it but how about the water resistance property of this biofilm?
- In figure 3, it seemed like that the FexOy NPs weren’t evenly fused into gelatin films. It looks more like these NPs formed aggregates among films.
Author Response
Reviewer: 2
Comments:
In this manuscript, the authors reported gelatin-based biofilms fused with different concentrations of FexOy-NPs and evaluated its performance in antioxidant and antimicrobial applications. Overall the work is well designed and the research topic is of interest to certain community. Whereas more solid data are needed to support the conclusion and a few issues should be addressed before acceptance of publication.
The authors acknowledge the reviewer’s comments. They have been valuable and very helpful for revising and improving the manuscript. We have carefully studied the comments and have made corrections. In addition, the quality of all the figures has been improved.
- There are a lot of antioxidant and antimicrobial biofilms that were either commercialized or reported. What is the strength of this gelatin-based FexOy-NPs biofilms compared with them? And the strength should be highlighted in the manuscript.
We appreciate the query of the reviewer. The relevance of biopolymer-based films was incorporated in the introduction section, including the biodegradability (L74-76) as well as the relevance of NPs as an antimicrobial agent and their incorporation in biofilms (L82-83). On the other hand, the authors apologise for not mentioning the inhibition of standard results used in this study (gallic acid). According to this comment, we have considered mentioning this fact in the Materials and methods (L171) and including standard results (L321-323) to be concluded at the end (L369-370). In addition, we have also considered including a comparison to other studies with appropriate references (L318-321).
- Biodegradability or the degradation behavior of this gelatin-based FexOy-NPs biofilms should be studied and presented.
The authors appreciate the reviewer’s comment and agree with him/her. Nevertheless, a proper biodegradability study cannot be carried out in such a short time. They will be performed in future works, as has been commented in the conclusion section (L375-377).
- Besides tables, many results of biofilms should be presented in the form of graphs or figures. E.g., mechanical parameters, antioxidant activity in table 1, and antibacterial activity in table 3.
Tables were used for their ease in comparing the specific values obtained for each system. The authors consider it important to know their exact values from graphics, which is why the tables have been maintained in the manuscript.
- There might be inconsistency between the color of pseudo-color scale bar and images in Figure 4. The authors should better double check.
We gratefully appreciate this observation. Figure 4 has been improved accordingly.
- Maybe I missed it but how about the water resistance property of this biofilm?
Water resistance of the films was assessed by water solubility (WS%) measurement (L129-135 and L192-212).
For the determination of this parameter, the samples (2 × 2 cm2) were firstly weighted (Wi) and, then, placed in an oven at 105 ºC for 24 h. Later, the samples were immersed in 50 mL of distilled water for 24 h. Finally, the films were taken out and redried at 105 ºC for 24 h to obtain the final dry weight (Wf). The weight loss or water solubility percentage (WS%) was calculated with Equation 1 (Andreuccetti et al. 2012; Hosseini et al. 2015; Mehmood et al. 2020):
(1)
Water solubility (WS) is a critical parameter for food packaging applications. In this sense, biofilms must be insoluble in water to improve water resistance and product safety (Hosseini et al. 2015; Mehmood et al. 2020). Table 1 shows the water solubility (WS) values of the different biofilms. As can be seen, the reference biofilm (neat gelatin) reached the highest WS value (80.9%). Higher WS of neat gelatin biofilms is due to the hydrophilic nature of gelatin (Voon et al. 2012). Thus, the incorporation of FexOy-NPs into the system improves its water resistance. Nevertheless, the increment in the FexOy-NPs concentration does not significantly improve their water resistance. Therefore, it could be concluded that the incorporation of NPs could decrease the solubility in water, regardless of the incorporated concentration. These results could be due to the formation of strong hydrogen bonds between the biopolymer chains and NPs, as has already been reported in previous works (Voon et al. 2012). Likewise, Wongphan et al. (2022) reported that the incorporation of enzymes into polymers could cause an interaction via hydrogen bonding that enhances the hydrophobic groups by reducing the contrast, which results in a decrease in solubility (Wongphan et al. 2022). These results are similar for GS and CS NPs.
REFERENCES
Andreuccetti, C.; Carvalho, R.A.; Galicia-García, T.; Martinez-Bustos, F.; González-Nuñez, R.; Grosso, C.R.F. Functional properties of gelatin-based films containing Yucca schidigera extract produced via casting, extrusion and blown extrusion processes: A preliminary study. J. Food Eng. 2012, 113, 33–40, doi:10.1016/j.jfoodeng.2012.05.031.
Hosseini, S.F.; Rezaei, M.; Zandi, M.; Farahmandghavi, F. Fabrication of bio-nanocomposite films based on fish gelatin reinforced with chitosan nanoparticles. Food Hydrocoll. 2015, 44, 172–182, doi:10.1016/j.foodhyd.2014.09.004.
Mehmood, Z.; Sadiq, M.B.; Khan, M.R. Gelatin nanocomposite films incorporated with magnetic iron oxide nanoparticles for shelf life extension of grapes. 2020, doi:10.1111/jfs.12814.
Voon, H.C.; Bhat, R.; Easa, A.M.; Liong, M.T.; Karim, A.A. Effect of Addition of Halloysite Nanoclay and SiO 2 Nanoparticles on Barrier and Mechanical Properties of Bovine Gelatin Films. Food Bioprocess Technol. 2012, 5, 1766–1774, doi:10.1007/s11947-010-0461-y.
Wongphan, P.; Khowthong, M.; Supatrawiporn, T.; Harnkarnsujarit, N. Novel edible starch films incorporating papain for meat tenderization. Food Packag. Shelf Life 2022, 31, 100787, doi:10.1016/j.fpsl.2021.100787.
- In figure 3, it seemed like that the FexOy-NPs weren’t evenly fused into gelatin films. It looks more like these NPs formed aggregates among films.
We appreciate this comment. We agree with this observation, and we have mentioned an explanation accordingly (L270-273). In fact, some authors (i.e. Phothisarattana et al. 2021) demonstrate that aggregation of nanoparticles in biodegradable polymers was possible with the increasing of nanofillers concentrations which may lead to changes in the mechanical and barrier properties of the films. Despite this, figure 4 shows a distribution of NPs throughout the surface of the film.
REFERENCES
Phothisarattana, D.; Wongphan, P.; Promhuad, K.; Promsorn, J. Films as Functional Active Packaging of Fresh Fruit. 2021.

Reviewer 3 Report
The paper entitled "Gelatin-based biofilms with FexOy-NPs incorporated for antioxidant and antimicrobial applications" by J. A. A. Abdullah, M. Jiménez-Rosado, A. Guerrero and A. Romero presents gelatin-based biofilms with embedded FexOy nanoparticles prepared by green synthesis (GS) and chemical synthesis (CS). The authors evaluated the physicochemical, mechanical, morphological and biological properties of resulting films.
The introduction is very well written and give a good insight into the topic. Overall the paper is clearly written, and the results well presented. The conclusions are well supported by experimental evidence.
Here are 3 minor points to fix:
1) Pg 1, Ln 29. The Reference [1] seems to be irrelevant since it deals with the SiO2 particle-size-effect on properties of nanoparticle assemblies. There is nothing about wrapping products or PP or PE in Reference [1]. The appropriate citation should be selected.
2) Pg 4, Ln 173. The “PBS” should be defined in the text in full as phosphate buffered saline.
3) The Supplementary Materials with images of inhibition area are missing, or I couldn’t find them in the submission. Please make sure it will be available.
The paper can be published in Materials if the following issue is appropriately addressed:
In section Materials, the authors claim that the NPs were synthesized according to a previous work that has not been published yet! The chemical (CS) and green (GS) methods of synthesis of NPs are quite essential for this paper. From the scientific point of view, it is not acceptable to omit the description of these synthetic procedures since it makes it impossible to repeat the experiments by readers. Please address this issue in the revised manuscript.
Author Response
Reviewer: 3
Comments:
The paper entitled "Gelatin-based biofilms with FexOy-NPs incorporated for antioxidant and antimicrobial applications" by J. A. A. Abdullah, M. Jiménez-Rosado, A. Guerrero and A. Romero presents gelatin-based biofilms with embedded FexOy nanoparticles prepared by green synthesis (GS) and chemical synthesis (CS). The authors evaluated the physicochemical, mechanical, morphological and biological properties of resulting films.
The introduction is very well written and give a good insight into the topic. Overall the paper is clearly written, and the results well presented. The conclusions are well supported by experimental evidence.
Here are 3 minor points to fix:
The authors are grateful for the valuable comments of the reviewer that contribute to improving the manuscript.
- Pg 1, Ln 29. The Reference [1] seems to be irrelevant since it deals with the SiO2 particle-size-effect on properties of nanoparticle assemblies. There is nothing about wrapping products or PP or PE in Reference [1]. The appropriate citation should be selected.
The authors gratefully appreciate this observation. According to this observation, appropriate citations has been included (L27 and L29).
- Pg 4, Ln 173. The “PBS” should be defined in the text in full as phosphate buffered saline.
The authors apologize for this mistake. The term has been correctly defined in the manuscript (L180).
- The Supplementary Materials with images of inhibition area are missing, or I couldn’t find them in the submission. Please make sure it will be available.
The authors apologise for this missing and gratefully appreciate this reminder. Accordingly, the Supplementary Materials would be available. These materials have been attached at the end.
- The paper can be published in Materials if the following issue is appropriately addressed:
In section Materials, the authors claim that the NPs were synthesized according to a previous work that has not been published yet! The chemical (CS) and green (GS) methods of synthesis of NPs are quite essential for this paper. From the scientific point of view, it is not acceptable to omit the description of these synthetic procedures since it makes it impossible to repeat the experiments by readers. Please address this issue in the revised manuscript.
The authors agree with this comment and regret not including this data in the previous version of this work. This information has been included in the new manuscript submission (L106-112).
SUPLEMENTARY MATERIALS
Gelatin-based biofilms with FexOy-NPs incorporated for antioxidant and antimicrobial applications
Johar Amir Ahmed Abdullah, Mercedes Jiménez-Rosado, Antonio Guerrero and Alberto Romero
Figure S1: Image of inhibition area over time of Neat gelatin-based biofilm without FexOy-NPs incorporated.
Figure S2: Image of inhibition area over time of gelatin-based biofilm with 1.0 % GS FexOy-NPs incorporated.
Figure S3: Image of inhibition area over time of gelatin-based biofilm with 1.0 % CS FexOy-NPs incorporated.

Round 2
Reviewer 1 Report
The authors revised the manuscript as recommend.
Reviewer 2 Report
The authors answered my questions and I have no further questions.